# Magnetite Nanoparticles in Magnetic Hyperthermia and Cancer Therapies: Challenges and Perspectives

**DOI:** 10.3390/nano12111807

**Published:** 2022-05-25

**Authors:** Agnieszka Włodarczyk, Szymon Gorgoń, Adrian Radoń, Karolina Bajdak-Rusinek

**Affiliations:** 1Department of Medical Genetics, Faculty of Medical Sciences in Katowice, Medical University of Silesia, Medyków 18, 40-752 Katowice, Poland; awlodarczyk@sum.edu.pl; 2Department of Surgical and Perioperative Sciences, Surgery, Umeå University, 901 87 Umeå, Sweden; szymon.gorgon@umu.se; 3Łukasiewicz Research Network—Institute of Non-Ferrous Metals, Sowinskiego 5 St., 44-100 Gliwice, Poland; adrian.radon@imn.lukasiewicz.gov.pl

**Keywords:** magnetite nanoparticles, hyperthermia, smart nanomedicine

## Abstract

Until now, strategies used to treat cancer are imperfect, and this generates the need to search for better and safer solutions. The biggest issue is the lack of selective interaction with neoplastic cells, which is associated with occurrence of side effects and significantly reduces the effectiveness of therapies. The use of nanoparticles in cancer can counteract these problems. One of the most promising nanoparticles is magnetite. Implementation of this nanoparticle can improve various treatment methods such as hyperthermia, targeted drug delivery, cancer genotherapy, and protein therapy. In the first case, its feature makes magnetite useful in magnetic hyperthermia. Interaction of magnetite with the altered magnetic field generates heat. This process results in raised temperature only in a desired part of a patient body. In other therapies, magnetite-based nanoparticles could serve as a carrier for various types of therapeutic load. The magnetic field would direct the drug-related magnetite nanoparticles to the pathological site. Therefore, this material can be used in protein and gene therapy or drug delivery. Since the magnetite nanoparticle can be used in various types of cancer treatment, they are extensively studied. Herein, we summarize the latest finding on the applicability of the magnetite nanoparticles, also addressing the most critical problems faced by smart nanomedicine in oncological therapies.

## 1. Introduction

Cancer is one of the most common causes of human death. In 2020 there were approximately 19.3 million cancer cases, and it is estimated that this number could increase to 28.4 million in 2040 [1]. Therefore, new cancer-treatment methods are tested and have been recently developed. These methods include improving well-established clinical methods such as chemotherapy, radiotherapy, or the development of less invasive surgery methods [2]. However, traditional methods still have a nonselective effect on neoplastic cells, thus interacting with healthy cells. This may result in limited effectiveness or adverse effect of treatment on a patient, even though these methods have the potential to kill cancer cells. Therefore, the development of new cancer treatments has focused on tumor-selective methods [3]. In recent years, selective inhibitors have been introduced into clinical use. They are used in targeted therapies against specific mutant signaling pathways [4]. Although inhibitors’ effectiveness has been proven, they are connected the with development of drug resistance by cancer cells [5]. Due to limitations and problems encountered in approved cancer therapies, there is a great need for new and better cancer treatment strategies.

One of the proposed alternative approaches in cancer treatment is magnetic hyperthermia [6]. Hyperthermia is already used to support the treatment of many cancers, e.g., head and neck, breast, cervical, sarcomas, and melanomas [2]. Although selective hyperthermia is not expected to be a standalone method, it could put selective pressure on cancer cells, increasing the therapeutic effect of primary cancer treatment. In the method, thermal energy is generated when magnetic nanoparticles are influenced by an alternating magnetic field [7]. Limiting the magnetic field gradient to a pathogenic site only allows increasing body temperature in the desired region of the body selectively.

Magnetic hyperthermia is based on the use of magnetic nanoparticles. Among the dozen or so magnetic nanoparticles, the most promising is magnetite. Magnetite nanoparticles are already in clinical use: they have found application as a diagnostic tool for contrasting [8]. In addition, there are plans to use magnetite in other ways in medicine. It has been proposed to use these nanoparticles in the separation of proteins. However, the wide range of magnetite properties makes them attractive nanoparticles that could be used in oncological therapies. Their potential in targeted drug delivery, gene, or protein therapy has been explored [8]. Increased interest in the topic is reflected in the number of published scientifical papers. An NCBI database search for the combined key words “magnetite” and “cancer” gives 1390 publications, including 7 clinical trials, published in the last 5 years [9].

In the presented work, we summarize the research and achievements to date in the field of medicine, biology, and chemistry that concentrate on therapeutic use of magnetite nanoparticles in oncology. In the presented context, magnetite research faces problems typical for translational medicine, i.e., in laboratory (both chemical and preclinical) and in clinics as well. Here, we describe the problems related to the medical use of magnetite nanoparticles and future research directions.

## 2. Nanoparticles

Nanoparticles (NPs) are defined as solid particles in the size range 10–1000 nm [10]; however, the definition introduced by the European Commission states that nanoparticles include materials where at least half of the particles are equal to or smaller than 100 nm [11]. In addition, they often exhibit new and distinct electrical, optical, magnetic, biological, and chemical properties [3,12]. Furthermore, nanoparticles can be synthesized from various materials, e.g., composite polymers, semiconductors, metals, or even lipids and proteins [13], and are characteristic of different shapes, including spheres, rods, or tubes [13]. Due to their various properties, there is growing interest in the use of nanoparticles to develop new materials of unusual and novel features [14,15]. Currently nanoparticles are tested in various fields such as catalysis [16,17], energy storage [18], energy conversion [19,20], and optoelectronic, for example, as light-emitting diodes [21]. In the case of the biomedical applications, NPs are tested not only as anticancer agents, which will be discussed later in this paper but also as antibacterial agents (for example in form of the layers on medical implants) [22,23], and sensitive tests to detect various diseases [24,25]. Generally, this wide application range is related to their unique properties and highly reactive surface. Unfortunately, their high surface energy can also result in poor stability [26].

Medicine is one of the branches of science where nanoparticles are expected to have the most significant impact. First and foremost, nanoparticles are planned to be used in so-called smart nanomedicine (Table 1). Here, the nanoparticle is coated using a polymer or other nanostructures such as metallic ones are deposited on their surface. It enables to attachment of bioactive substances such as cytotoxic or therapeutic drugs [27]. Targeting and activating nanoparticles occurs under various factors, including magnetic field, light waves, ultrasounds, and internal, including pH, temperature, redox potential, or enzymes [28]. This procedure allows for the obtaining of the desired therapeutic effect at the pathogenic site only. Another promising use of nanoscale objects is a theranostic nanoparticle, i.e., a particle that can be used in diagnostics and simultaneously therapeutic agents [29,30,31].

Currently, they are used in cancer medicine, mainly in diagnostics. In addition, nanoparticles serve as a stabilizing factor for therapeutics when used in therapies. According to the authors’ knowledge, about 10 types of nanoparticle-based therapies have been approved by the U.S. Food and Drug Administration (FDA) and European Medicines Agency(EMA) [32].

### Iron-Oxide Nanoparticles

Nanoparticles that have been used in medicine can be divided into three groups: metal, nonmetallic, and composite nanoparticles. The most commonly used are metal and metal-oxide nanoparticles [3]. Because of biocompatibility and unique magnetic properties, the most widely studied are ferrites, especially superparamagnetic iron-oxide nanoparticles (SPIONs). Recently, iron-oxide nanoparticles have attracted much consideration due to their unique properties, such as superparamagnetism, surface-to-volume ratio, greater surface area, and easy separation methodology. Superparamagnetic nanoparticles are susceptible to the effects of a magnetic field; however, when the external magnetic field is lost, they become demagnetized. This feature, together with good colloidal stability, is clinically important. It minimizes the risk of nanoparticles’ aggregation in the blood [48]. Additionally, the biodegradability of SPIONs seems to be one of the most important features for clinical use [49,50,51].

Within the group can be found such iron oxides as maghemite (γ-Fe_2_O_3_), hematite (α-Fe_2_O_3_), and magnetite (Fe_3_O_4_) [52]. From a medical point of view, magnetite seems to be the most promising SPION. It is due to its unique magnetic, catalytic, and biochemical properties [53]. Fe_3_O_4_ nanoparticles have a large surface area needed for adsorption and immobilization of molecules or drugs. They can be easily modified to improve absorption [54]. Moreover, their size and shape can be easily controlled by using different synthesis methods [55]. It is possible to synthesize multifunctional platforms using Fe_3_O_4_ nanoparticles, in which other compounds cover magnetite nanoparticles, such as SiO_2_, or even by the biodegradable polymers [56,57,58].

Therapeutic magnetite nanoparticles usually consist of three essential components: magnetite core, which serves as a carrier for therapeutic agents; a coating on the iron-oxide nanoparticle, which promotes beneficial interactions in the body; and a therapeutic load that performs a designated function in vivo (Figure 1). In addition, a particle may consist of a ligand that recognizes specific receptors on the outer surface of cancer cells [59]. Interestingly, these structures can also be modified, and magnetite nanoparticles with or without coating can be deposited on the other nanostructures such as layered double hydroxides (LDHs). Interestingly, Fe_3_O_4_ nanoparticles can also be covered by LDHs to form nanostructures with different morphology and properties [60]. This flexibility of synthesis of nanostructured multifunctional platforms results in the high interest in applying magnetite in nanomedicine. Fedorenko et al. [61] have shown that the ultrafine iron-oxide nanoparticles can be covered by the silica layer and this composite can be doped by the Gd(III) complexes to improve their MRI dual-contrast ability. Formation of the core-shell structure results in much higher stability in dispersion, which is connected with the high electrokinetic potential (about −78 mW). Moreover, not only the formation of this type of nanostructures can change the properties of iron oxides. As was recently presented, the modification of the surface by various surfactants can highly modify the magnetic properties and also the transverse relaxation rate of water molecules in the aqueous colloids of hydrophilic magnetic nanoparticles [62,63].

## 3. Hyperthermia

Hyperthermia is defined as an increase in body temperature by ≥1 °C [64] or as an increase in temperature to 39–45 °C [65]. There are four types of hypothermia: fever, movement-related hyperthermia, inadequate temperature drainage hyperthermia, and pathological or medication-induced hyperthermia [66]. Healthy human somatic cells, except neurones, can survive at a temperature of 44 °C for at least 1 h [67,68]. Thus, hyperthermia does not damage healthy tissues unless the temperature exceeds the above value. On the other hand, hyperthermia in the temperature range 45–50 °C causes thermal ablation, i.e., necrosis through dehydration, protein denaturation, and cell-membrane damage [65,69]. Although hyperthermia can be dangerous to the organism, it can be of clinical use, especially when directed against specific cells or tissue. Medical hyperthermia is classified according to the size of the area where the temperature is raised. Thus, hyperthermia is divided into local-regional and whole-body hyperthermia [2].

Increasing temperature in a limited-body region allows maximizing therapeutic effect in the appropriate parts of the body while minimizing undesirable effects of heating in other body regions [70]. There are several possible therapeutically methods of increasing temperature in a specific body fragment. One of them is magnetic hyperthermia [71]. In the method, the alternating magnetic field generates electromagnetic radiation. This energy is absorbed by SPION and converted into thermal energy [72,73]. As a result, the surrounding tissue heats up. A similar method of increasing body temperature is photothermic. Here, instead of the alternating magnetic field, a laser is used as a source of electromagnetic radiation [13,73,74]. Therefore, laser-induced temperature elevation may be a more effective method of reaching the desired temperature than magnetically induced hyperthermia [74]. First of all, light with partially transparent tissues would reduce the main drawback of magnetic field-induced hyperthermia, i.e., low tissue penetration [73]. Both photothermia and magnetic hyperthermia have been intensively studied as potential cancer treatments [75,76,77].

### 3.1. Hyperthermia in Oncology

In the 18th century, it was noticed that tumors shrank in patients with fever. Since then, hyperthermia has been intensively studied as a potential cancer treatment [78]. However, the first attempts to use hyperthermia in the clinic were made in the 20th century [79]. At the beginning of the century, in vitro and in vivo studies showed that neoplastic cells do not tolerate increased temperature as well as healthy cells [80,81]. Furthermore, toxicity tests in dogs, sheep, and pigs showed that progressive necrosis of normal and neoplastic tissues occurred at temperatures above 45 °C. However, internal heat dissipation occurred as the normal tissues approached this temperature. This phenomenon resulted, in healthy tissues maintaining the temperature below 45 °C, with practically no damage to normal organs, tissue, or skin. However, most solid tumors did not have this adaptability and heated up to 50 °C [82].

So far, many studies have been carried out to confirm the effects of hyperthermia [2]. Initially, hyperthermia was planned to be used as a standalone therapy [79]. However, hyperthermia was studied mainly as a support to other therapies. This approach includes a combination of hyperthermia and chemotherapy [73,83], radiotherapy [71,84,85] or with both [86,87]. Results showed that increasing body temperature increases the effectiveness of therapies [71,73,83,88,89], which was also estimated and confirmed by many clinical studies [2,90]. 

Currently, oncological hyperthermia is used primarily as an adjunctive method in treating many cancer types, along with chemotherapy or radiotherapy [2,71,78]. In this case, the patient’s body temperature is usually raised to around 40–43 °C [88,89]. In addition, hyperthermia is also used as an independent therapy for ablation of individual neoplastic lesions, where a temperature above 50 °C is used [71].

Although many studies show the positive effect of hyperthermia in cancer treatment, this method is still controversial, and the results of clinical trials have been criticized [2,91]. In addition, some problems make it difficult to assess the effect of hyperthermia on tumors in a clinical setting. The main problem is the lack of a precise oncological definition of hyperthermia [2]. In addition, it makes it difficult to standardize the methodology. Moreover, the unresolved technical problems with hyperthermia are limiting factors in the wide use of hyperthermia as a cancer-treatment strategy (see Section 5). Due to the above problems, hyperthermia is a rarely used strategy in oncology even when it is clinically approved for use in some cancer therapies [2,92].

#### 3.1.1. How Hyperthermia Kills Neoplastic Cells?

Although many studies confirmed the effectiveness of hyperthermia, the exact mechanism of its action is not fully understood [2,91]. The main problem in pinpointing the exact mechanism is fact that many changes occur in a cell at high temperatures [91]. So far, several explanations have been given. First of all, the effect of hyperthermia on tumors may occur at the cellular level. In vitro studies have shown that temperatures above 41 °C block the DNA double-break repair mechanism [2]. During the S-phase of cell division, cells tend to be less susceptible to the effects of therapeutics. This changes when the temperature increases and the cells show increased sensitivity to the therapies. Some cytostatics interact with cells during the S-phase of division so that hyperthermia could enhance the effect of these drugs [2]. The fast rate of cell divisions and hypoxia characteristic of cancer makes cancer cells more susceptible to hyperthermy [93]. It has also been shown that the combination of gold nanowires with hyperthermia can damage the cytoskeleton and affect the migration of cancer cells [94] or synergistically support the inactivation of the complex of oncogenic proteins PML/RARα [95,96]. In the case of tumors with a high level of well-known oncogenic proteins p53 and Ki-67, the addition of hyperthermia to chemical treatment improves the effectiveness of the therapy [97]. Other studies have shown that hyperthermia can induce apoptosis through p53-dependent and -independent mechanisms [98].

Moreover, hyperthermia can help overcome the problem of drug resistance by modulation of different pathways [88]. However, more studies are needed. For example, results show that hyperthermia can act positively and negatively in developing drug resistance and can induce or help overcome resistance [88].

However, hyperthermia can affect tumors by modulation of blood flow [91]. In healthy tissue, blood vessels are hierarchically connected into efficient networks of arteries, capillaries, and veins. On the other hand, cancerous blood vessels are chaotic, leaky, and inefficient [99]. Moderate hyperthermia (T < 42 °C) increases blood flow in the neoplastic tissue, increasing the number of cytostatic drugs in the neoplastic tissue, and thus has an adjunctive effect on chemotherapy. In addition, blood flow increases the availability of oxygen to cancer cells. Neoplastic-cell hypoxia is related to radioresistance. Thus, increased blood flow also makes the tumor more susceptible to radiotherapy [78].

Further temperature increase (above 42 °C) is connected with additional effects. High temperature damages chaotically connected blood vessels and reduces blood perfusion in the cancerous tissue. This causes local hypoxia and acidification of the tumor, and eventually necrosis. However, the increase does not strongly affect healthy tissue [91]. In addition, hyperthermia can stimulate the immune system in the cancer environment by programming the immune system to recognize cancer cells [93,100,101].

#### 3.1.2. Magnetite Nanoparticles in Magnetic Hyperthermia

There are several problems associated with hyperthermia nowadays. The main problems are related to nonselective heating up of tissues and not reaching a high enough temperature [102,103]. Therefore, attempts have been made to overcome these problems. One of the most promising solutions is based on magnetic nanoparticles. They can strongly absorb microwave radiation [104] and transform it into heat. With precise delivery of microwave radiation to the desired place, localization of hyperthermia to the pathological site is possible. Among the magnetic nanoparticles, due to the magnetic properties and efficiency in energy dissipation (in this case, the conversion of magnetic energy into thermal energy), the most promising seems to be magnetite-based (Fe_3_O_4_) [105]. There are also other significant features of Fe_3_O_4_ nanoparticles that can be used in treatment. These include targeting them to neoplastic cells through increased permeability and retention [106] and enabling highly specific active and passive targeting [107]. In addition, magnetite is easy to manipulate to obtain the desired effect. The heating of the magnetite depends on the frequency and amplitude of the applied magnetic field and the size, morphology, and composition of the magnetic nanoparticles [103,108]. In addition to microwave radiation (magnetic hyperthermia), magnetite can be induced with the use of a laser (photothermal) [109].

The use of ferrites, such as magnetite, in cancer treatment has been under investigation since the 1950s [7,110]. Since then, much laboratory research has been conducted on ferrites in cancer treatment [31,111]. The magnetite nanoparticles can be synthesized in different shapes; their size and surface functionalization can be easily controlled and modified. Accordingly, the possibility of their application in the magnetic hyperthermia field is widely tested. Recent research on the applicability of magnetite nanoparticles with controlled size and shape as hyperthermia agents is summarized in Table 2. As can be seen, according to the lack of a standardized measurement procedure, it is difficult to compare results obtained for the same material: Fe_3_O_4_ nanoparticles. The specific absorption rate (SAR), defined as the power transformed into heat per unit of mass of nanoparticles, strongly depends on the alternative current (AC) fields parameters (frequency and field strength) and the magnetic fluid concentrations. However, the size and shape of magnetite also play a crucial role in the efficiency of magnetic hyperthermia. For example, magnetite nanorings are characterized by higher SAR values than nanotubes, while spherical-shaped magnetite nanoparticles can achieve SAR values of about 11 W/g for two times lower concentrations than magnetite-truncated octahedrons in similar AC magnetic fields parameters (see Table 2).

The effect of ferrites has been studied in in vitro cell cultures, as well as in in vivo models, both in the case of magnetically induced hyperthermia [8,124] and photothermia [74,125]. Moreover, the hypothermic effect generated by ferrites was tested on organoids [126]. Several clinical trials have been performed using magnetite-based hyperthermia [127]. The first clinical use of interstitial hyperthermia using magnetite nanoparticles was reported in 2005 [128]. The procedure was performed on the prostate gland of a 67-year-old patient. In a technique also known as magnetic fluid hyperthermia, magnetite nanoparticles were injected into the target area and selectively heated by an externally applied alternating magnetic field [128]. According to the authors’ knowledge, so far, magnetite nanoparticles have been used therapeutically in procedures performed on patients who have prostate cancer [128,129], melanoma [130], and glioblastoma [131]. Although many studies have proven the positive effect of ferrites (especially magnetites in cancer treatment), it has not been possible to develop an appropriate methodology to allow for the broader use of magnetic hyperthermia. Many preclinical studies have been conducted, but few reach the third and final phase of clinical trials [8]. The year 2021 may be the turning point. NanoTherm^®^ (treatment of glioblastoma) completed clinical phase 2a, and the NoCanTher project has started final clinical trials (treatment of advanced pancreatic ductal adenocarcinoma) [132].

## 4. Use of Magnetite Nanoparticles in Cancer Therapies

Besides the application of magnetite in magnetic hyperthermia, there are other proposed applications of these nanoparticles in cancer therapies. The nanoparticle’s properties allow use for targeting cancer. Interaction between the magnetite particle and magnetic field is the backbone of targeting. Magnetite-carrying drugs can be pulled to the desired location in the body by the magnetic field. Thus, the magnetite nanoparticles described here are carriers for therapeutics. Several studies have been made to test the possible use of magnetite in the controlled delivery of drugs or gene and protein therapy [8].

### 4.1. Targeted Delivery of Medicinal Substances

Nowadays, the main problem of cancer treatments is nonselectivity toward cancer cells; interactions with healthy tissue cause numerous side effects [133]. In chemotherapy, cytostatic drugs block cell proliferation. Cancer cells are characteristic of rapid cell division and are strongly affected by cytostatic drugs. However, healthy cells also divide, and the therapeutics interact with them. The side effects of chemotherapy include hair loss, nausea and vomiting, associated loss of appetite, or bone-marrow disorders.

Additionally, new disease entities may develop even long after the end of chemotherapy, e.g., pulmonary tissue fibrosis, infertility, cardiac arrhythmias, and peripheral neuropathies [134,135]. Precise delivery of medicinal substances to the pathological site would reduce side effects. In addition, targeted drug delivery allows the use of a higher concentration of drugs than in the case of other therapies [136]. In traditional treatment strategies, high drug concentrations could be toxic and cause severe damage to healthy tissue [137]. Injecting therapeutic close to its target could help reach a high concentration; however, it may be dangerous for a patient [136].

#### 4.1.1. Nanopharmaceuticals

The magnetic particle can serve as a carrier in controlled drug delivery [8,133,138]. The implementation of nanoscale pharmaceutical products, i.e., nanopharmaceuticals, allows for the improvement of the physicochemical properties of not only existing drugs, but also new molecular units with biological activity [139]. In nanopharmaceuticals, active ingredients (APIs) are loaded into a nanocarrier to improve their solubility, extend the half-life, improve pharmacokinetic properties, obtain a modified release profile, and reduce their toxicity [139].

#### 4.1.2. Nanopharmaceuticals in Magnetic Field for Targeted Drug Delivery

Therapeutics can then be directed to the desired location through a magnetic field [137]. The magnetic nanoparticle–drug complex is delivered by manipulating an external magnetic field using electromagnetic coils [137] to various types of permanent magnets [140] (Figure 2). In addition, nanoparticles can be functionalized, e.g., with folic-acid groups that interact with receptors specific for some tumors to increase specificity [141]. The potential of these delivery systems for anticancer drugs has been shown in several studies. For example, magnetite nanoparticles with attached carboplatin particles showed a high rate of inhibition of ovarian tumors without significant toxicity to healthy organs [142]. Similarly, promising research has been carried out on a multifunctional nanoplatform delivering cisplatin to cancer cells [143]. The platform consisted of biodegradable PLGA surface-functionalized by chitosan and polyvinyl alcohol with additional SPIONs. Tests showed increased release time of cisplatin to more than 10 days and confirmed the possibility of using nanoparticles also as an imaging agent. Another study has shown that drug-coated polymer nanospheres and nanocapsules enhance intracellular anticancer effects [144]. The method of drug delivery into the body was also developed using multilayer microcapsules composed of poly (allylamine) and poly (sodium 4-styrene sulfonate). This method has been tested in in vitro and in vivo conditions [145].

However, the magnetic field could be used in drug delivery another way [146]. It was proposed to enclose drugs in a polymer–magnetic beads microparticle complex. In case of the absence of a magnetic field, the drug is slowly realized from a carrier. The drug realizes rate increases when the magnetic field is applied because pores are compressed [146]. 

### 4.2. Cancer Genotherapy

Gene therapy aims to modify genetic material or correct genetic errors that cause diseases. It can be performed by injecting foreign genetic material into the body (in vivo gene therapy) [147,148] and if the modified cell is outside the patient’s body, then reintroduced into the body (ex vivo gene therapy) [147,148,149]. Gene therapy can be used in therapies of many diseases; however, here we focus only on its use in oncology.

Several proposed oncological therapies are based on the modification of nucleic acids [148]. These include the suicide gene, immunization gene, oncogene silencing, tumor-suppressor-gene replacement, angiogenesis-targeting therapy, and others [148]. In addition, the cancer vaccine, which is considered one of the most promising oncological therapies, is also based on DNA modifications [150]. However, despite numerous studies on the described techniques, the effectiveness of the proposed strategies still does not provide the full potential of gene therapy in oncology [148]. The main problem is the lack of an effective system to deliver genetic material to desired cells and tissue.

#### 4.2.1. Nanoparticles in Genotherapy

The introduction of therapeutic nucleic acids into the target cell can be done only using appropriate vectors. So far, several methods of delivering genetic material to the patient’s body have been proposed. Among them are distinguished methods based on viral vectors and physical and chemical methods [147,151]. The use of the vectors proposed so far in cancer gene therapy is associated with immunogenicity, limited genetic burden, and cancer risk due to the insertion of a therapeutic load in the vicinity of genes responsible for controlling cell growth [152]. Consequently, there has been engagement in potentially using nanoparticles as vectors in cancer-gene therapy. Due to the simple synthesis and low cytotoxicity, nanoparticles seem to be ideal candidates.

Additionally, easy surface modification of the nanoparticles enables immune-system-response modulation [153]. Nanoparticles provide a new approach to gene vectors because they can wrap or adsorb nucleic acids on their surface. Specific targeting molecules, such as a monoclonal antibody, are then coupled to the particle surface, and the nanoparticles may be endocytosed or phagocytosed by cells, and the encapsulated therapeutic agents may be effective in a cell [49]. Some unique nanomaterials have magnetic, optical, and thermal properties that enable the delivery and controlled release of target genes [154]. By coating nanoparticles with hydrophilic polymers, it is possible to obtain a low level of adsorption to blood proteins, which helps to avoid phagocytosis [155]. Due to their low immunogenicity and easy cell-membrane penetration, nanoparticles are probably one of the most promising vectorization strategies for delivery, as reflected in the number of clinical trials proposed using nanovectors for cancer treatment [148]. Nanoparticles can be used with viral vectors, but as nonviral vectors [156].

#### 4.2.2. Magnetite in Genotherapy

Nanovectors based on magnetite nanoparticles are also considered a potential vector in gene therapies. Initially, magnetite was tested with an adenoviral vector [157]. One of the limitations of adenoviral vectors is the low cell penetration due to the short time of exposure of the cell to the virus. Studies have shown that binding a vector to a magnetite nanoparticle can extend the exposure of the cell to the vector and thus also increase cell penetration [158]. However, magnetite has been studied as the primary vector. Magnetic transfection, or magnetofection, delivers genetic material to the cell using an external magnetic field [8,159]. Magnetite nanoparticles (with nucleic acids attached) are pulled by magnetic force to the tissue. In addition, magnetic force helps penetrate the cell membrane [160]. The presence of nanoparticles near the tissue also increases the absorption of the genetic material through endocytosis [161]. It was found that such use of the magnetic field can significantly increase the efficiency of transfection of foreign genetic material into the cell [162]. Naked magnetite particles or particles coated with, e.g., polyethyleneimine (PEI), to which plasmids, short hairpin RNA (shRNA), or antisense oligonucleotides are attached [8]. Numerous in vitro studies confirm the effectiveness of magnetofection based on SPIONs, both in animals and plants [162,163,164,165,166,167,168]. However, there are no clinically approved methods of delivering genetic material based on nanoparticles, and their use is limited to laboratory tests [148,156].

### 4.3. Protein Therapy

In cancer therapies, the goal of protein therapy is to deliver a sufficient dose of the therapeutic protein to cancer cells [59]. It is considered one of the most direct and safe cancer therapies described so far [59]. Protein therapy may include blocking a cell-surface receptor involved in carcinogenesis, intracellular delivery of proteins involved in cell signaling, inhibiting growth and invasion of cancer cells, or inducing apoptosis [59,169]. Capsule methods have been developed to maintain the function of the therapeutic protein introduced into the body [169]. However, the targeted delivery and tracking of encapsulated therapeutic proteins to the desired cells remain challenging. It turns out that the use of nanoparticles can result in the effective delivery of the therapeutic protein to cancer cells while maintaining their enzymatic activity [169,170]. Nanoparticles can protect the proteins carried against the activity of proteases or facilitate their penetration into the cell [59].

According to the authors’ knowledge, magnetite nanoparticles were not clinically used in protein therapy, and magnetite was tested in in vivo conditions. In the mouse model, the therapy based on magnetite nanoparticles and the combination with chlorotoxin inhibited the invasiveness of glioblastoma [171]. In addition, enhancement of the drug effect in target cells using nanoparticles was also described [171]. This is because the therapeutic protein can be distributed explicitly on the surface of the nanoparticles. Another example is the use of iron oxide to deliver an antibody. The antibody effectively inhibited the mutant EGFR receptor [172]. In addition, the encapsulation of cytochrome c in the nanoparticle cavities induced apoptosis after delivery to tumor cells [169].

## 5. Challenges in the Development of Magnetite-Based Therapies

The development of new cancer therapies, especially selective strategies, is connected with several well-known problems that make it challenging to develop a uniform methodology. These include heterogeneity of tumors, differences between the human and mouse models used in in vivo studies and differences between neoplastic lesions [15]. In addition, there are several problems connected with the use of magnetite nanoparticles in magnetic hyperthermia that require careful further studies. These unsolved problems are related to hyperthermia, general problems with nanoparticles, and technological and methodological aspects of the synthesis and use of magnetite.

In recent years, hyperthermia has been used to support radiotherapy, chemotherapy, and immunotherapy, increasing their effectiveness [78,173]. However, hyperthermia still lacks a uniform definition in oncology. This makes the standardization of methodology impossible. It results in variation between studies, which gives contradictory results [2]. It also makes it impossible to calculate and optimize the appropriate dose of energy [2]. The exact mechanism of hyperthermia-mediated cancer treatment has to be further studied. Especially one aspect of hyperthermia in oncology needs to look deeper. Studies on the effects of hyperthermia on drug resistance show opposite results and suggest that, in some cases, hyperthermia may stimulate the development of resistance [88].

Although the properties of nanoparticles show promise from a clinical point of view, they require further research. Unfortunately, nanotechnology develops faster than legislature and it creates problem of lack of standardization of nomenclature and test methodology. There is great need for guidelines for safety assessment of nanomaterials [174]. Detailed understanding of how nanoparticles interact with cells, proteins, hormones, or immune factors is fundamental to their commercial and clinical application [104]. First of all, it is unknown what happens with nanoparticles when they are metabolized in the body. Do nanoparticles, or their metabolites, accumulate in cells or organs, thereby triggering intracellular changes, inflammatory responses, and the development of oxidative stress? It has been suggested that nanoparticles can be transported in the body by the neuronal route [175], which may contribute to the degeneration of the nervous system. Increased ROS production, induced by nanoparticles, may damage organelles and contribute to the increased aggregation of proteins involved in the pathogenesis of Alzheimer’s disease and Parkinson’s disease [176,177,178]. Some studies suggest that nanoparticles may affect the patient’s endothelium and contribute to the development of cardiovascular diseases [179].

Moreover, they can enter the blood cells, which can damage morphotic elements of the blood. There are also reports of an allergic reaction following the use of nanoparticles [180,181]. Nanoparticles can also absorb biological agents such as bacterial endotoxins on their surface, which may also have an unknown risk [182].

Cytotoxicity of nanomaterials depends on the concentration, proportion, porosity, aggregation tendency, a chemical affinity for biological structures, chemical surface reactivity and even the shape, size of nanoparticles, and environmental conditions [183,184]. So far, many studies have been conducted on the cytotoxicity of magnetite [31,111,138,185,186,187,188]. First of all, it has been noticed that the previously mentioned particle size influences the cytotoxicity of magnetite. When using chicken macrophages, it was found that smaller magnetite particles (60, 120, and 250 nm were used) were more toxic to these cells [188]. No significant toxicity of particles with a size of 30 μm was detected on HeLa cells and postnatal human fibroblasts [138]. Interestingly, it has been shown that magnetite can be toxic to fibroblasts; however, the same concentration of nanoparticles does not affect leukocytes in the tested parameters [187]. Farcas et al. [111] found that small magnetite particles (up to 30 nm) are not toxic to human keranocytes and primary endodermal melanocytes but may be toxic to melanoma cells, both in the human and murine cell lines [111]. However, modifications of magnetite nanoparticles can eliminate problem of cytotoxicity [189]. An example is the study in which the cytotoxicity of nanoparticles based on the magnetite core was reduced when used with an oleic-acid shell [189].

The synthesis of magnetite is a leading technical problem and requires improvement [190]. It is challenging to obtain magnetite particles of appropriate and the same size with known synthesis methods. For extensive use of magnetite, it is also needed to reduce the cost of synthesis. Large-scale synthesis or the laboriousness of coating nanoparticles are also limiting factors [132,190]. Methods of magnetite synthesis based on the co-precipitation method allow nanoparticles with high efficiency; however, the obtained structures are strongly agglomerated [191].

On the other hand, methods based on the high-temperature decomposition of organic precursors allow for the obtaining of monodisperse nanoparticles, albeit with low efficiency compared to the reaction time [191,192]. The most frequently used method of synthesizing nanoparticles with high application potential (e.g., magnetic hyperthermia and targeted therapies) is the polyol method [193]. However, a much lower efficiency characterizes it than in the case of the coprecipitation method. It is also much more challenging to control the nucleation and growth of nanoparticles than reactions in high-boiling liquids and using, e.g., oleic acid or stearic acid [194,195]. The major problem in targeted drug delivery based on magnetite is the limited packaging capacity of the nanoparticles. Here, using magnetite together with other materials could be an option. The most promising seems to be a combination of nanoparticles and hydrogels as a drug carrier [196]. 

Another noteworthy obstacle connected with a large-scale synthesis of therapeutic is fact that this is clinically useful only when the quality of obtained therapeutic is unchanged between batches of manufactured product. Therefore, a reproducible, scalable-production-method nanoparticle-based therapeutic must be developed and validated [139]. Here, good manufacturing practices of nanotherapeutics should ensure that guidelines recommended by agencies that control the authorization and licensing of the pharmaceutics [139].

The clinical methodology also needs improvement. The main problem limiting the effectiveness of magnetic hyperthermia is low penetration of the magnetic field, i.e., up to 5 cm into the body due to the loss of the magnetic field gradient [133]. As a result, magnetite can only be activated in subcutaneous tissues. There are attempts to use magnetic implants that generate sufficient magnetic field strength to attract magnetic nanoparticles carrying drugs [133]. However, even with implants, an external magnetic field is required to obtain the desired effect [197]. In addition, if a patient has metal implants, e.g., of the hip joint, or a pacemaker or implantable heart defibrillator, the procedure cannot be performed [198].

Additionally, in the case of prostate cancer, the temperatures typically reached inside the tumor were generally in the range of 40–41 °C. Only a few of these studies have reached maximum intraprostate temperatures of 42.5 °C or exceeded the critical temperature of 43.0 °C. The temperature distribution in the prostate gland is heterogeneous, so reliable temperature calculations during prostate-cancer hyperthermia are challenging to obtain. Due to local differences in tissue properties, excessive temperature rise can occur, resulting in burns, blisters, pain, thrombosis, bleeding, and other undesirable effects [89]. Therefore, the region affected by the temperature rise must be strictly limited to the desired tumor volume. In the case of local-regional hyperthermia, it is necessary to diagnose all tumors. If not all lesions are recognized, unrecognized lesions are left untreated [2]. This phenomenon is fundamental when the magnetic field is intended to activate drug release from nanoparticle carriers.

## 6. Conclusions

Tumor-selective therapies should be the main oncological strategies in the future. Nanoparticles’ properties could be helpful in the development of such methods. Magnetite seems to be one of the most promising nanoparticles in oncology. This is mainly because the nanoparticle can be used in different ways, and all of the proposed magnetite-based strategies selectively interact with tumors. The therapeutic application of magnetite nanoparticles has been under investigation for many years with no success. Most of the tests ended at the preclinical phase [8]. There are still several obstacles to the wide use of magnetite in medicine. However, nanoparticles are expected to revolutionize medicine and pharmacy, resulting in nanomedicine being an intensively developing branch of medicine. Many problems connected with safe use and nanoparticles synthesis will probably be solved in the near future. These problems can be connected with a few aspects. First of all, the hyperthermia measurements should be standardized. The presented research in this field currently cannot be easily compared to decide which way (such as surface, shape, or size modifications) should be tested in the future to develop highly biocompatible particles characterized by high magnetically induced hyperthermia effect. Secondly, new synthesis methods of iron oxides should be developed. These methods should be characterized by repeatability, relatively high yield, and most critically, scalability. The currently tested methods based on the coprecipitation and high-temperature degradation processes result in the synthesis of agglomerated nanoparticles or synthesis of nanoparticles with low yield but characterized by uniform size and shape. Above that, further research should also concentrate on biocompatibility, elimination from the body, and the long-term effects of nanoparticles on cells. There are many contradicting literature reports on these aspects, for example, in the case of the potential use of cobalt ferrite nanoparticles.

## Figures and Tables

**Figure 1 nanomaterials-12-01807-f001:**
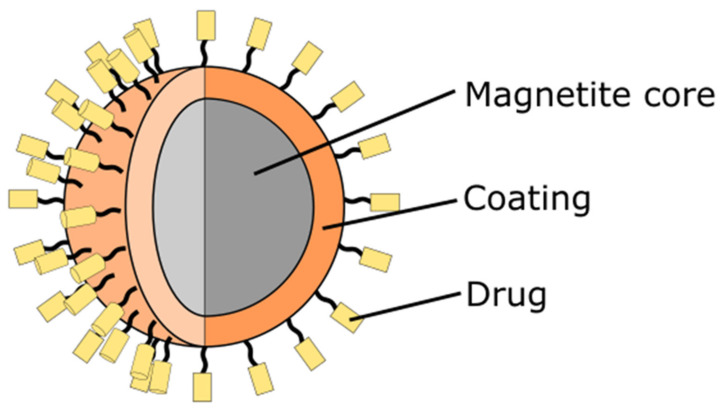
Schematic representation of a therapeutic magnetite-based nanoparticle. In typical magnetite-based nanoparticle, core of the nanoparticle is made of magnetite. The core is covered with a coating substance. On the surface, therapeutic load is attached.

**Figure 2 nanomaterials-12-01807-f002:**
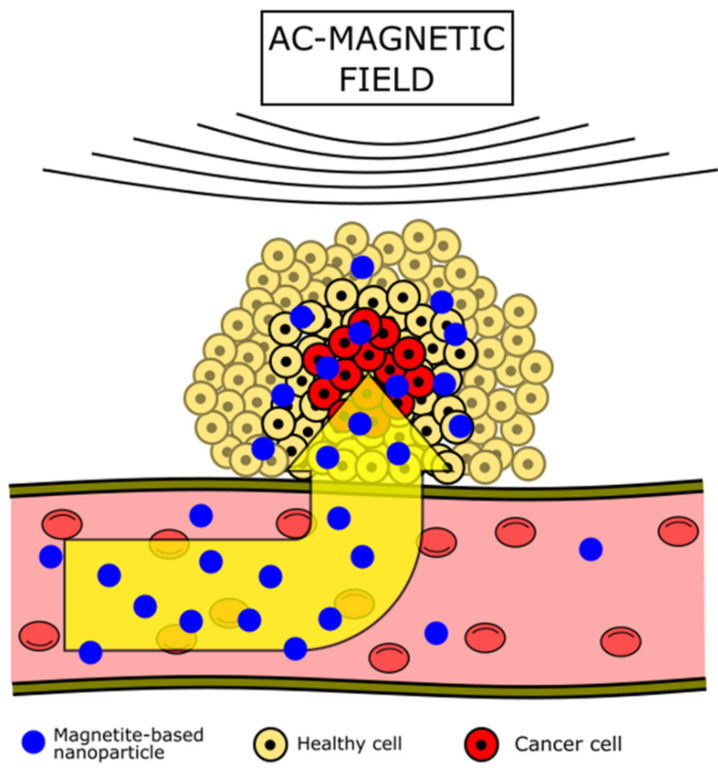
Magnetite-based drug delivery occurs under the influence of an external AC magnetic field. Nanoparticles are pulled from a blood vessel to a pathological site and migrate toward the source of altering magnetic field. The pulled-out nanoparticles are concentrated in a tumorous site with minimal interaction with healthy cells. Then, release of the therapeutics load occurs.

**Table 1 nanomaterials-12-01807-t001:** The potential applications of nanoparticles in smart nanomedicine.

Group	Nanoparticle	Drugs, Functionalizing Agents, or Bioactive Molecules	Size (nm)	Shape	Nanoparticle Role	Application Type	Reference
**Metallic**	Au	hesperidin	10–50 ^1^	spherical	Drug-delivery system	antioxidant agent; protective agent against DNA damage from H_2_O_2_, anticancer drug-delivery system for breast cancer therapy	[33]
	AgCu	sodium citrate, mercapto-propionic acid	4–32 ^2^	spherical	anticancer agent	selective toxicity against MCF-7 breast cancer cells	[34]
	Pt	doxorubicin, fucoidan	33 ± 3.4 ^3^	coral-like	photothermal agent, drug-delivery system	nanoplatforms for synergetic biological-thermo-chemo trimodal treatment of MCF-7 breast cancer cells	[35]
	Au@Pt	methoxy-PEG-thiol	20–30	dendritic	photothermal agent	photothermal therapy under 808 nm light irradiation	[36]
	Pd	polyvinylpyrrolidone	9–15	spherical	anticancer agent	ROS generation and cleaving of the mitochondrial membrane	[37]
**Nonmetallic**	MnAl layered double hydroxide	Fluorouracil	34.9 and 24.7 ^4^	nanosheets	Drug-delivery system, MRI contrast	anticancer drug-delivery system combined with MRI	[38]
	ZnO	quercetin	21–39	hexagonal	Drug-delivery system	pH responsive targeted nano-drug-delivery system	[39]
	Fe_3_O_4_	BSA protein, glutaric acid	4.5 ± 0.1 ^5^; 98 ^6^	spherical	magnetic hyperthermia agent	hyperthermia therapy	[40]
	CuS	Doxorubicin, DGM ^7^, peptides ^8^	22 ^9^; 10 ^10^; 151.5 ± 2.2 ^11^	spherical	cross-linker, photothermal agent	pH and redox responsive photochemotherapy	[41]
	graphene quantum dots	TAT peptides, FA-PEG-NH_2_ ^12^	5	spherical	anticancer agent	selectively damaging of the cancer cells DNA	[42]
**Composite**	Au@SiO_2_	poly (*N*-isopropylacrylamide-co-acrylic acid); indocyanine green	59 ± 3.6 ^13^; 9.3 ± 1.2 ^14^	nanorods	Drug-delivery system, photothermal agent	photodynamic/photothermal tumor therapy by combining the effect of Au NPs with ICG under irradiation of 808 nm laser	[43]
	Fe_3_O_4_@SiO_2_	lactoferrin, doxorubicin	119 ^15^	spherical	Drug-delivery system, magnetic hyperthermia, and photothermal agent	chemo-magnetic field-photothermal breast cancer therapy	[44]
	silica-carbon nano-onion	fucoidan, doxorubicin, HM30181A	~50	nano-onions	Drug-delivery system, photothermal agent	anticancer drug releasing combined with the binding of P-glycoprotein pumps and photothermal effect	[45]
	AlMg layered double hydroxide-Fe_3_O_4_	hyaluronic acid, doxorubicin	3.2 ± 0.2 ^16^ 68.8 ± 10.7 ^17^	disk-shaped	Drug-delivery system, MRI contrast,	*T1*-weighted MR imaging and chemotherapy by targeting cancer cells overexpressing CD44 receptors	[46]
	Ag@SiO_2_ Ag@SiO_2_@Ag	n.a.	77 ± 13 ^18^; 83 ± 18 ^19^	core-shell spherical	bioimaging agent, photothermal agents	bioimaging and photothermal therapy using 400 nm laser	[47]

^1^ for hesperidin-loaded Au NPs, ^2^ depending on Ag: Cu ratio, ^3^ for fucoidan coated Pt NPs, ^4^ measured at two different dimensions, ^5^ for pure Fe_3_O_4_ NPs. ^6^ hydrodynamic diameter for Pro-Glu-Fe_3_O_4_ NPs (Pro—BSA protein, Glu—glutaric acid), ^7^ poly(ε-caprolactone)-ss-poly(2-(diisopropylamino)ethyl methacrylate/glycidyl methacrylate/2-methylacrylloxyethyl phosphorylcholine, ^8^ cyclo(AGA-d-PC) and (YGRKKKRRQRRRC) peptides, ^9^ CuS size determined based on dynamic light scattering, ^10^ CuS size determined based on the TEM measurements, ^11^ sizes for cross-linked micelles, ^12^ folic acid-linked polyethylene glycol, ^13^ Au nanorod length, ^14^ Au nanorod width, ^15^ for lactoferrin loaded composite ^16^ for pure Fe_3_O_4_ NPs, ^17^ for composite, ^18^ for nanoparticles with 5 nm Ag seeds on the surface, ^19^ for nanoparticles with 12 nm Ag NPs on the surface.

**Table 2 nanomaterials-12-01807-t002:** Comparison of specific absorption rate (SAR) of magnetite nanoparticles with various shapes (C—magnetic fluid concentration, f—AC magnetic field frequency, H—magnetic field strength).

Shape	Nanoparticle Size (nm)	C (mg/mL)	f (kHz)	H (Oe)	SAR (Wg)	Reference
truncated octahedron	10 ± 3	10	112	250	8.4	[112]
nanorods	50–70 ^1^, 8–10 ^2^	5	184	261	69	[113]
				395	154	
				522	246	
nanorings	67 ± 21 ^3^, 41 ± 18 ^4^, 93 ± 27 ^5^, 26 ± 12 ^6^	0.6	300	560	~1000	[114]
nanotubes	470 ± 45 ^1^, 110 ± 20 ^4^, 170 ± 20 ^5^, 55 ± 5 ^6^	1	300	400	80	[115]
nanorings	55 ± 5 ^1^, 55 ± 5 ^4^, 110 ± 15 ^5^, 55 ± 5 ^6^	1	300	400	100	
porous	<100	5	220	117	480	[116]
porous/hollow	350	5	220	117	140	
nanodiscs	26 ^7^, 225 ^2^	0.1 ^8^	488	400	420	[117]
				600.6	5000	
ellipsoidal	250 ^1^, 50 ^2^	5	765	300	759^9^	[118]
polyhedral	5.1	10	100	300	10.1	[119]
	14.3				42.2	
	53.1				71.6	
	225.6				19.1	
spherical	11.4 ± 3.2	5	109.8	240	11	[120]
spherical	10 ± 2	0.5	1001.1	125.4	432.1 ^9^	[121]
			751.5	137.6	330.8 ^9^	
			522.2	193.4	189.3 ^9^	
elongated	12 ± 3	2 ^8^	435	193.5	900 ± 22 ^9^	[122]
cuboidal	15 ± 2				658 ± 53 ^9^	
cuboidal	80.78 ± 3.37	5	315	300	160 ± 1.5	[123]
				400	166 ± 1.4	
				500	209 ± 1.6	
				600	317 ± 1.2	
				700	430 ± 1.4	
				800	1036 ± 1.5	

^1^ length, ^2^ diameter, ^3^ height, ^4^ inner diameter, ^5^ external diameter, ^6^ wall thickness, ^7^ thickness, ^8^ Fe concentration in dispersion, ^9^ W/g_Fe_.

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
