# Peer review of "Magnetite Nanoparticles in Magnetic Hyperthermia and Cancer Therapies: Challenges and Perspectives"

_nanomaterials, 2022, doi:10.3390/nano12111807_

Round 1

Reviewer 1 Report

The review by Agnieszka Włodarczyk et al. is aimed at providing fresh insight into the field of nanomaterials utilized for magnetic hyperthermia and cancer therapies. This comprehensive review contains 156 references mainly composed of references belong to last five-year period. Actually, 86 papers issued during the period of 2017-2021, which is 55% of total amount of items in the reference list, were thoroughly picked from the huge literature massive to be collected in this review. Due to the fast and extensive development in the field in is generally recommended to use latest articles and to avoid outdated information, which is usually covered by previous reviews. In this regard this review fully agrees with this requirement. The only recommendation is that the authors are encouraged to refer following fresh paper 10.1016/j.colsurfa.2018.09.044; 10.1016/j.colsurfa.2013.12.009 to the paragraph 2.1. Iron oxide nanoparticle. The structure of the review is good and friendly to the reader. Please add contents to facilitate orientation in the text. In some cases like p53 and Ki-67 (Page 6 of the MS) and other narrow professional terms would be perfect if authors let the readers of Nanomaterials know what they are? Was oxygen meant instead oxide in the following expression “the availability of oxide to cancer cells”? Concluding, this paper is well-written, precise and interesting to the reader. It brings substantial input to scientific literature with deep overview in the field.

Author Response

Reviewer #1:

1. Reviewer #1 suggested to add two papers to the paragraph 2.1. Iron oxide nanoparticle.

The Authors want to thank you to the Reviewer for his time and deep insight into the review article. According to the suggestion, the mentioned papers were introduced and discussed in paragraph “Iron oxide nanoparticles”.

2. Reviewer #1 asked to add context to several specialistic names where the case of proteins p53 and Ki-67 was underline.

We aim to have our manuscript reader-friendly and straightforward. Therefore, we want to especially thank for this comment. We have corrected manuscript and added descriptions in case we felt such are needed.

The function of the p53 tumor suppressor gene is known as the "guardian of the genome". In case, if the DNA damage is too severe to be repaired, p53 induces apoptosis that eliminates cells from the tissue. This suggests that high level of the protein is beneficial to a cell and of anti-cancer activity. However, p53 is often mutated in cancer ang high levels of the protein are connected with some tumors. To avoid misunderstanding and to do not confuse a reader, we have kept such biological descriptions as short as possible.

3. Reviewer #1 asked a specific question: ”Was oxygen meant instead oxide in the following expression “the availability of oxide to cancer cells”?”

We are sorry for this mistake and authors want to thank the Reviewer for pin-pointing the mistake. It should be oxygen and the sentence was corrected according to this.

Reviewer 2 Report

The novelty character of this Review respect to the others present in literature should be better marked. 

The section on nanomaterials should be described giving major information.The authors should describe the application of nanomaterials in different fields and also mention aspects linked to regulation and safety and related references should be added such as:

Souto et al. Nanopharmaceutics: Part I-Clinical Trials Legislation and Good Manufacturing Practices (GMP) of Nanotherapeutics in the EU. Pharmaceutics. 2020;12(2):146. Published 2020 Feb 11. doi:10.3390/pharmaceutics12020146

Yeung et la. 2020. Big impact of nanoparticles: analysis of the most cited nanopharmaceuticals and nanonutraceuticals research. Current Research in Biotechnology Volume 2, November 2020, Pages 53-63

The subparagraph 2.1. Iron oxide nanoparticles shoudl be better introduced and linked to preview text.

More details and information in Figure 1 should be given.

The part 3.2 should be a subparagraph of 3.1 or included in it

Table 2 should give a more representative picture.

More details of Figure 2 should be given,

The Conclusion should be reflects the paper content: more considerations should be added as well as limits, advantages and future applications.

A section Methodology including bibliographic research criteria should be inserted, including a graphical rapresentation.

Author Response

Reviewer #2:

1. ”The novelty character of this Review respect to the others present in literature should be better marked.”

We added information that the review combine knowledge from different fields i.e., medicine, biology and chemistry. According to the author knowledge, it would be the only one review of this type that has been recently published.

2. ”The section on nanomaterials should be described giving major information. The authors should describe the application of nanomaterials in different fields and also mention aspects linked to regulation and safety and related references should be added...”

We agree with this remark, accordingly mentioned section was extended. Some of the recent papers related to the applicability of nanoparticles in various fields and their limitations were introduced and discussed in new manuscript version.

3. The subparagraph 2.1. Iron oxide nanoparticles should be better introduced and linked to preview text.

According to this remark, we have added additional information in introduction to the section. We hope it will provide better understanding why iron oxide nanoparticles are used so widely.

4. More details and information in Figure 1 should be given.

According to this remark, more descriptive legend for the Figure 1 was added.

5. The part 3.2 should be a subparagraph of 3.1 or included in it

According to this remark, section 3.2 is subparagraph of the section 3.1 in the corrected manuscript.

6. Table 2 should give a more representative picture.

According to this remark, the Table 2 was extended.

7. More details of Figure 2 should be given

According to this remark, more descriptive legend for the Figure 2 was added.

8. The Conclusion should be reflects the paper content: more considerations should be added as well as limits, advantages and future applications.

According to this remark, the Conclusion section was rewritten to show the potential research ways, which should be tested in near future to improve the applicability of nanoparticles in biomedical field.

9. A section Methodology including bibliographic research criteria should be inserted, including a graphical rapresentation.

Beside background for an appropriate section, during preparation of the manuscript authors mainly focused on review of literature no older than 5 years. This was also underline by Reviewer #1. We checked all accessible publications in pursuit of relevant information. Many publications do not clearly state that magnetite nanoparticles were used in a given study. Such publications were excluded.

In our opinion, adding a section concentrated on Methodology could blurry main aim of the manuscript. However, information about how many papers concentrated on the topic have been recently published, could show a reader increased interest in the topic. Therefore, relevant information was added.